# Watching Too Much Television is Good: Self-Supervised Audio-Visual Representation Learning from Movies and TV Shows

## Abstract

The abundance and ease of utilizing sound, along with the fact that auditory clues reveal so much about what happens in the scene, make the audio-visual space a perfectly intuitive choice for self-supervised representation learning. However, the current literature suggests that training on *uncurated* data yields considerably poorer representations compared to the *curated* alternatives collected in supervised manner, and the gap only narrows when the volume of data significantly increases. Furthermore, the quality of learned representations is known to be heavily influenced by the size and taxonomy of the curated datasets used for self-supervised training. This begs the question of whether we are celebrating too early on catching up with supervised learning when our self-supervised efforts still rely almost exclusively on curated data. In this paper, we study the efficacy of learning from Movies and TV Shows as forms of uncurated data for audio-visual self-supervised learning. We demonstrate that a simple model based on contrastive learning, trained on a collection of movies and TV shows, not only dramatically outperforms more complex methods which are trained on orders of magnitudes larger uncurated datasets, but also performs very competitively with the state-of-the-art that learns from large-scale curated data. We identify that audiovisual patterns like the appearance of the main character or prominent scenes and mise-en-scène which frequently occur through the whole duration of a movie, lead to an overabundance of easy negative instances in the contrastive learning formulation. Capitalizing on such observation, we propose a hierarchical sampling policy, which despite its simplicity, effectively improves the performance, particularly when learning from TV shows which naturally face less semantic diversity.

## 1   Introduction

Recently, there has been tremendous progress in self-supervised learning from still images, where the standard supervised training has been outperformed in a variety of image-related tasks [7, 8, 15, 29]. The appeal of detaching representation learning from human annotations is rooted not only in the non-trivial challenges of scaling-up the labeling process, but also in the ill-defined task of determining a proper taxonomy with generalization power and transferability. Both challenges only exacerbate as we move from images to videos, where the notion of time is involved and the complexity of visual concepts increases. Simply considering the number of training instances or even the cardinality of the label set is not sufficient to conclude if one large-scale supervised dataset is more suitable than another for transfer learning in video classification tasks [20]. That is, the abundance of attention which video self-supervised learning has lately received is only to be expected. While many research efforts in this area extend the contributions made initially in the image domain to the video domain,

Submitted to 35th Conference on Neural Information Processing Systems (NeurIPS 2021). Do not distribute.

others, including our work, have explored harnessing additional modalities such as audio or text for multi-modal self-supervised learning [2, 3, 4, 22, 27, 31, 37, 36, 39].

From the current state-of-the-art one makes two major conclusions. First, the quality of learned representations, evaluated by fine-tuning on downstream tasks, is heavily influenced by the size and taxonomy of the pretraining datasets [2, 3, 39]. Second, an *uncurated* pretraining dataset yields considerably poorer representations compared to a *curated* one and the gap only narrows when the total amount of pretraining data significantly increases [3]. Curated data refers to likes of *supervised* large-scale action recognition and audio classification datasets such as Kinetics [6], IG-Kinetics [12], AudioSet [11], and YouTube-8M [1]. While the human-annotated labels are not accessed for self-supervised pretraining, videos being trimmed and from a label set of limited cardinality with biased sampling distribution[1] implicitly acts as a sort of supervision. On the other hand, an *uncurated* data refers to likes of IG-Random[3], simply a body of unlabeled videos collected blindly with none of the aforementioned careful human-involvements. That being said, we know that something as simple as having access to a clean object-centric training data, like Imagenet, can be indirectly exploited by contrastive self-supervised learning in image domain to obtain additional performance gain [41] on the downstream tasks which exhibit similar properties. The analogous to it of course are the well trimmed closed-set curated datasets which are being extensively used in the literature for video self-supervised pretraining, while downstream evaluations focus on benchmarks with similar characteristics. Our work aims at comprehensively exploring the efficacy of learning from Movies and TV Shows, as forms of uncurated data, for audio-visual self-supervised learning.

Many of us can relate to an experience in movie theaters when the sound of the engine, first perceived by our left ear, is gradually heard more by the right ear as a car moves from the left side of the screen to the right side. Another example is a scene in which an object, like a helicopter, approaches the camera from distance and eventually flies over it. In this case, the perceived sound not only changes in loudness but also transitions from front to back, in concert with the visuals, giving the audience a more realistic feeling as if they are indeed positioned behind the camera. Besides, with art being inherently novel, two movies even if they share genres or revolve around similar story lines often deliver quite different experiences and portray distinct visuals, thanks to the extremely artist-driven creative process behind such productions. We hypothesize that the aforementioned high audio fidelity, and inherent semantic diversity characterize long-form content[2] as potentially a very rich source for self-supervised multi-modal representation learning. It is worth emphasizing that in spirit of uncurated data, we not only blindly sample from a large collection of movies and TV shows when constructing our pretraining dataset, but also perform ablation studies on the effect of genre distribution, the closest we have to taxonomy in the curated datasets, confirming that the quality of learned representations is agnostic with respect to such statistics.

To the best of our knowledge, we are the first to solely rely on uncurated data and study the efficacy of self-supervised multi-modal representation learning from movies and TV shows. Despite meaningful domain gap between our pretraining data and the space of downstream tasks, we obtain representations which are very competitive with those learned from curated datasets. This is particularly important as we follow a much simpler modeling approach in comparison with the state-of-the art.

## 2   Related Work

Self-supervised learning techniques define *pretext* tasks, mostly inspired by the natural structures in the data, in order to generate supervisory signals for training. Despite the plethora of proposed *pretext* tasks in the literature, these approaches can be coarsely divided into two groups, namely *pretext learning*, and *pretext-invariant* methods. Approaches which fall in the former bucket, usually apply a form of transform, randomly drawn from a parametric family, to the input data then optimize for predicting the parameters of the chosen transformation. Predicting the relative position of image patches [9], solving jigsaw puzzles [33], estimating artificial rotations [13], colorization [50], context encoders learned through inpainting [38], and learning by counting scale and split invariant visual primitives [34], are among many methods which belong to this category. Similar techniques have been extended from images to videos [10, 21, 24, 25, 30, 46, 48, 49], where in addition to the spatial context, the temporal domain, and the arrow of time have been heavily exploited. In contrast,

---

[1]the associated taxonomy has similarities with those of downstream benchmarks [3]

[2]alternatively referring to movies and TV shows

*pretext-invariant* methods [5, 7, 8, 15, 18, 17, 29, 35, 39, 44] are built on the concept of maximizing mutual information across augmented versions of a single instance, and are mostly formulated as contrastive learning. In other words, a pretext is used to generate different views of a single input for which the learning algorithm aims to maximize the intra-instance similarity, across variety of transformations. Our work falls within this category, however we function in a multi-modal realm employing both audio and video.

Earlier works which harnessed audio and video for representation learning, have leveraged audio-visual temporal synchronization [22, 36], correspondence [4], and cross-modal clustering [3, 37]. The work by Patrick *et al.*[39] proposes a generalized data transformation in order to unify a variety of audio-visual self-supervised pretext tasks through a noise contrastive formulation. This work is close to ours in choice of objective function and data type, yet we employ no augmentation (except *modality projection* in the terminology of [39]), and solely focus on capitalizing the advantages of learning from long-form content. Morgado *et al.*[31] show that cross-modal discrimination is important for learning good audio and video representations, something which was also pointed out earlier in a clustering framework [3]. Beyond that, [31] generalizes the notion of instance-level positive and negative examples by exploring cross-modal agreement where multiple instances are grouped together as positives by measuring their similarity in both the video and audio feature spaces. While we also adopt a cross-modal noise contrastive estimation loss, we stick with the vanilla version, instance-level positive and negatives, and do not use any memory bank feature representations. Finally, Alayrac *et al.*[2] recently proposed a multi-modal versatile network capable of simultaneously learning from audio, video and text. Building on the intuition that different modalities are of different semantic granularity, audio and video are first compared in a fine-grained space while text is compared with the aforementioned modalities in a lower dimensional coarse-grained space. In our experiments, we compare with a variant of [2] where only audio and video modalities are utilized.

## 3   Approach

**Notations and Architecture.**  Our pretraining dataset is denoted by $\mathcal{X} = \{\mathcal{X}_n | n \in [1 \cdots N]\}$, where $\mathcal{X}_n = \{x_{n,m} | m \in [1 \cdots M_n]\}$ contains $M_n$ non-overlapping audiovisual snippets which are temporally segmented from the duration of the $n^{th}$ long-form content in the dataset. Each snippet includes both audio and video modalities, formally $x_{n,m} = (a_{n,m}, v_{n,m})$, where $a_{n,m} \in \mathbb{R}^{1 \times P \times Q}$ and $v_{n,m} \in \mathbb{R}^{3 \times T \times H \times W}$. $T$, $H$, and $W$ denote the number of frames, height and width of the video, while $P$, and $Q$ respectively stand for the number of mel filters, and audio frames. Video and audio are processed through 18-layers deep R(2+1)D [45] and ResNet [16] architectures, respectively referred to as $f : \mathbb{R}^3 \to \mathbb{R}^{d_f}$ and $g : \mathbb{R}^1 \to \mathbb{R}^{d_g}$. Inspired by [7], we use *projection heads*, $h_f : \mathbb{R}^{d_f} \to \mathbb{R}^d$ and $h_g : \mathbb{R}^{d_g} \to \mathbb{R}^d$, to map corresponding representations into a common $d$-dimensional space before computing the contrastive loss. The shallow architecture of $h_f$ and $h_g$ consists of two convolution layers, separated by Batch Normalization [19] and ReLU [32], followed by global average pooling. Once self-supervised pretraining finished, we discard the projection heads and fine-tune $f$ and $g$ for respective downstream tasks.

**Loss Function.**  With a slight abuse of notation[3], $\mathcal{B} = \{x_i = (a_i, v_i) | i \in [1 \cdots B]\}$ represents a minibatch of size $B$, where video and audio modalities associated with the $i^{th}$ sample, $x_i$, are denoted by $v_i$ and $a_i$. We use $z_v^i = h_f(f(v_i))$ and $z_a^i = h_g(g(a_i))$ to represent the associated embeddings generated by projection heads, and optimize the noise-contrastive loss [14] shown in 1 in order to maximize the symmetric joint probability between audio and video. For the $i^{th}$ element in the minibatch, $(z_v^i, z_a^i)$ serves as the positive pair, while assuming negative pairs for both modalities, $\mathcal{N}_i = \{(z_v^i, z_a^j), (z_v^j, z_a^i) | j \in [1 \cdots B], i \neq j\}$ constitutes the set of negative pairs.

$$\mathcal{L} = -\sum_{i=1}^{B} \log \left( \frac{e^{(z_v^i)^\intercal (z_a^i)}}{e^{(z_v^i)^\intercal (z_a^i)} + \sum_{(z_v', z_a') \in \mathcal{N}_i} e^{(z_v')^\intercal (z_a')}} \right) \tag{1}$$

Most of the previous works [2, 31, 39] normalize the embeddings before computing the contrastive loss and employ a temperature hyper-parameter, often denoted by $\tau$ as in [2, 31], to control the

---

[3]$i$ enumerates elements in the minibatch

smoothness for the distribution of pairwise similarities. In contrast, we have chosen to operate in an unnormalized embedding space. Besides the obvious benefit of eliminating the need for tuning $\tau$, we empirically show that such decision does not affect the quality of the learned representations.

**Sampling Policy.** Contrastive loss function shown in Equation 1 is computed over $B$ training instances, each in form of an audiovisual snippet. A naive sampling policy may ignore the fact that snippets comprising the pretraining dataset are in fact temporal segments that were trimmed from longer-form contents, *i.e.* movies and TV shows. Such an assumption treats our training data as independent and identically distributed random variables from $\bigcup_{n=1}^{N} \mathcal{X}_n$, which constitutes the default sampling policy that is commonly used in the general deep learning literature. However, in reality, commonalities and correlations do exist along the temporal axis of a movie or TV show, things like audio mastering artifacts, frequent appearance of the main character's face and voice, thematic music, repetitive scenes and mise-en-scène[4], all of which contribute to breaking the previously discussed i.i.d assumption. This is even more pronounced when we deal with multiple episodes of the same TV show appearing in the pretraining dataset[5]. Note that, sampling from no video data is going to be i.i.d but in this case the temporal correlations extend for much longer given our entities are movies and TV shows. Thus, it is more accurate to think of $\mathcal{X}$ having multiple underlying domains, oriented towards exclusive properties which different long-form contents are characterized by. We hypothesize that during training, model gradually discovers such patterns of commonalities, which are not semantically valuable, and latches onto those to quickly minimize Equation 1 leading to poor generalization[6]. The reason being $B \ll N$, hence for $n \sim \mathbb{U}(1, N)$ and $m \neq m'$, $\mathsf{P}(x_{n,m} \in \mathcal{B} \wedge x_{n,m'} \in \mathcal{B})$ is negligible. In other words, the set of negative pairs in Equation 1 mainly includes pairs for which audio and video come from two different movies or TV shows, thus due to the aforementioned artifacts behave as easy negatives.

In order to quantitatively measure our hypothesis, we define different distributions, shown in Equation 2, over the space of audio-visual similarity. $\mathcal{S}^+$ indicates the space of correct matches, *i.e.* where audio and video correspond to the same snippet. $\mathcal{S}^-$ indicates the space where audio and video do not correspond yet belong to the same movie or TV show. Finally, $\mathcal{S}^{\neq}$ indicates the space in which audio and video are sampled from two distinct long-form content, hence naturally do not correspond.

$$(z_v^{n,m})^\mathsf{T}(z_a^{n',m'}) \sim \begin{cases} \mathcal{S}^+, & \text{if } n = n' \wedge m = m' \\ \mathcal{S}^-, & \text{if } n = n' \wedge m \neq m' \\ \mathcal{S}^{\neq}, & \text{if } n \neq n' \wedge \forall (m, m') \end{cases} \tag{2}$$

With that, and KL denoting Kullback–Leibler divergence, $\mathsf{KL}(\mathcal{S}^- \parallel \mathcal{S}^+)$ measures the expected difference between positive and negative pairs within the same movie or TV show. Ideally, this should increase as the training progresses, since the model gradually learns audio-video correspondence by minimizing Equation 1. Meanwhile, the i.i.d assumption suggests $\mathsf{KL}(\mathcal{S}^- \parallel \mathcal{S}^+) \simeq \mathsf{KL}(\mathcal{S}^{\neq} \parallel \mathcal{S}^+)$ and $\mathsf{KL}(\mathcal{S}^- \parallel \mathcal{S}^{\neq}) \simeq 0$, yet as we empirically illustrate later, $\mathsf{KL}(\mathcal{S}^- \parallel \mathcal{S}^+) < \mathsf{KL}(\mathcal{S}^{\neq} \parallel \mathcal{S}^+)$ and $\mathsf{KL}(\mathcal{S}^- \parallel \mathcal{S}^{\neq})$ is rather large, indicating that, upon convergence and on a held-out set, model has a harder time pushing apart negative pairs when audio and video come from the same underlying long-form content. Next, we explain how a simple alternative policy which samples $k$ snippets from each long-form content effectively reduces both of the *discrepancy measures*, referring to $\mathsf{KL}(\mathcal{S}^- \parallel \mathcal{S}^{\neq})$ and $\mathsf{KL}(\mathcal{S}^{\neq} \parallel \mathcal{S}^+) - \mathsf{KL}(\mathcal{S}^- \parallel \mathcal{S}^+)$, while yielding better generalization on a range of downstream tasks.

To ameliorate the aforementioned optimization challenge, we take a hierarchical approach. In particular, we first uniformly sample a long-form content, $n \sim \mathbb{U}(1, N)$, and then draw $k$ distinct snippets from $\mathcal{X}_n$, creating $\{x_{n,m} | m \in \mathcal{M}_n\}$, where $\mathcal{M}_n \subset [1 \cdots M_n]$ and $|\mathcal{M}_n| = k$. This ensures that for $x_i \in \mathcal{B}$, $\mathcal{N}_i$ always includes $2k - 2$ pairs sampled from the same movie or TV show to which $x_i$ belongs. By putting constraints on $\mathcal{M}_n$, specifically how temporally far from each other the $k$ samples are drawn, we may go one step further and to some extent control the audiovisual similarity between snippets. This serves as an additional nob to tune for hard negative sampling. The intuition is that, the larger narrative of a professionally made movie or TV show is composed of shorter units called *scene*. Each scene comprises a complete event, action, or block of storytelling and

---

[4]collectively referred to as *content-exclusive artifacts*

[5]alternatively think of it as a very long movie created by stitching different episodes together

[6]refer to supplemental material for illustrations of training loss

normally takes place in one location and deals with one action. That is, if our samples are temporally close, it is more likely for corresponding snippets to be highly correlated and/or look/sound alike. $k \leq \max[\mathcal{M}_n] - \min[\mathcal{M}_n] + 1 \leq w \leq M_n$ defines the bounds on our sampling policy, where $w$, standing for a sampling *window*, determines the farthest two out of $k$ samples drawn from $\mathcal{X}_n$ can be. Accordingly, $w = k$ represents the case where all $k$ samples are temporally adjacent, hence the expected audiovisual similarity is maximized due to temporal continuity in content. We show that having such level of hard negatives, even with a small $k$, prevents proper training and results in performance degradation. On the other hand, $w = M_n$ indicates random sampling where no temporal constraint is imposed on $\mathcal{M}_n$, thus samples are less likely to be drawn from adjacent time-stamps. In this case, expected audiovisual similarity (*i.e.* hardness of negative pairs) is mainly derived from global content-exclusive artifacts like, color palette, frequent appearance of the main character's face and voice, repetitive scenes, and etc. The rest of the spectrum provides middle grounds where two samples drawn from $\mathcal{X}_n$ can at most be $w + 1$ snippets apart, something reminiscent of temporal locality. Our sampling policy can be easily implemented in a few lines of `Python`. Please refer to supplemental material for further details.

# 4 Experiments

## 4.1 Experimental Setup

**Datasets and Reproducibility.** We use full-length movies and episodes of TV shows for self-supervised pretraining. Titles are randomly chosen from a large collection spanning over a variety of genres, namely Drama, Comedy, Action, Horror, Thriller, Sci-Fi and Romance. All audio is in English language. Our Movie dataset, consists of 3.6K films with an average duration of 105 minutes. Our TV dataset includes 9.2K episodes from a total of 581 shows with an average duration of 42 minutes per episode. Each of our datasets comprises 0.7 years worth of uncurated audiovisual content, which is significantly smaller than IG-Random [3] with variants at 5 and 21 years. Scaling up our pretraining datasets to volumes comparable to the IG-Random [3] while possible is non-trivial and demands dramatically larger compute resources for training, something which we currently cannot afford. Given that we cannot publicly release our dataset due to copyright reasons, we acknowledge that it is not possible for other research groups to fully reproduce our results. However, we intend to make available the pretrained models and hope that research community finds them, along with the other contributions of this work, of value whether within the context of self-supervised learning or adoption for various downstream tasks. We would like to emphasize that similar limitations have precedents in multiple earlier works including but not limited to [3, 12, 26, 43]. To evaluate the efficacy of self-supervised audio-visual representation learning from movies and TV shows, we follow recent works [3, 39, 31, 2] and benchmark UCF101[42] and HMDB51[23] for action recognition, along with ESC50[40] for audio classification. Results for the ablation studies are reported on the split-1 of the corresponding datasets. Following the standard protocol, we report the average performance over all splits when we are comparing with the state-of-the-art.

**Pretraining.** Unless mentioned otherwise, we use video snippets with 16 frames at 5 fps. For data augmentation, we resize the shorter side to 190 pixels, then randomly crop them into $158 \times 158$ pixels. As for sound, we compute mel spectrogram from the raw audio at 48K sample rate using 96 mel filters and an FFT window of 2048, while the number of samples between successive frames is set to 512. For data augmentation, we randomly drop out up to 25% from either temporal or frequency axis of the 2-D mel spectrogram image. Training uses a batch size of 512 and takes on average 42 hours on 8 NVIDIA A100 GPUs. The dimension of audio-video joint embedding space, $d$, is set to 512.

**Downstream Evaluation.** For training on UCF101 [42] and HMDB51 [23], we use video clips that are 32 frames long at 10 fps. Unless mentioned otherwise, these clips are randomly chosen from the duration of the video instances. A scale jittering range of [181, 226] pixels is used and we randomly crop the video into $158 \times 158$ pixels. Furthermore, random horizontal flipping and color jittering are employed. During inference, 10 temporal clips are uniformly sampled where each is spatially cropped in 3 ways (left, center, right) resulting in a total of 30 views. We then average the model predictions across these 30 views and report top-1 classification accuracy. For training on ESC50 [40], we use 3-seconds clips which are randomly chosen from the duration of the audio instances and apply time and frequency masking to spectrogram images for data augmentation. The maximum possible length of the mask is 50% of the corresponding axis. We do not use any scale jittering or random cropping on the spectrograms. During inference, 10 temporal clips are uniformly sampled

238 and we average the model predictions across these 10 views and report top-1 classification accuracy.
239 For further implementation details, please refer to the supplemental material.

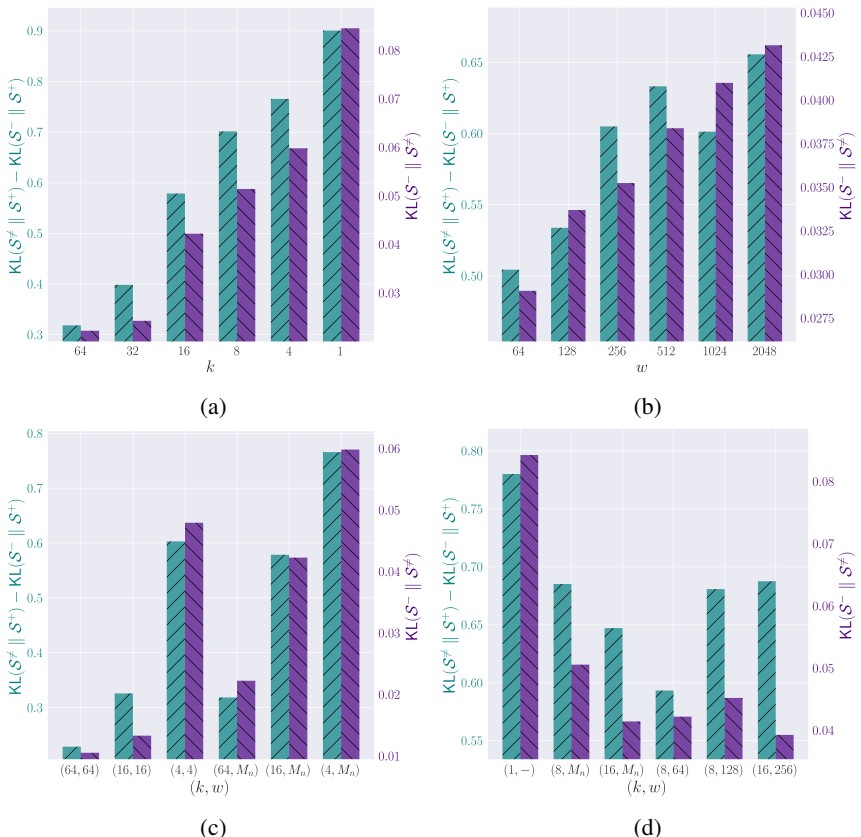

(a)                                  (b)

(c)                                  (d)

Figure 1: Ablation study of the proposed sampling policy on reducing the discrepancy measures.

## 4.2 Ablation Study

241 In the following, we discuss multiple ablation studies to
242 assess our main hypothesis that, a hierarchical sampling
243 policy, as described in Section 3, enables better repre-
244 sentations to be learned by increasing the portion of hard
245 negative pairs which the contrastive loss function observes.
246 Here, pretraining uses 90% of either Movie or TV dataset,
247 while the remaining 10% constitute a held-out validation
248 set[7] on which we report the discrepancy measures.

249 **Sample size ($k$)** Figure 1a illustrates that compared to
250 the baseline sampling denoted by $k = 1$, our approach
251 ($k > 1$) effectively shrinks the gap between $\mathcal{S}^-$ and $\mathcal{S}^{\neq}$
252 when measured either directly or against $\mathcal{S}^+$. Its pattern
253 of behavior also perfectly follows our earlier intuition (ref.
254 Section 3). In particular, given a fixed minibatch budget,
255 a larger $k$ favors more training instances to be sampled
256 from fewer number of long-form contents. That increases
257 the portion of hard negative pairs, thus pushes the con-
258 trastive loss to more aggressively separate mismatched

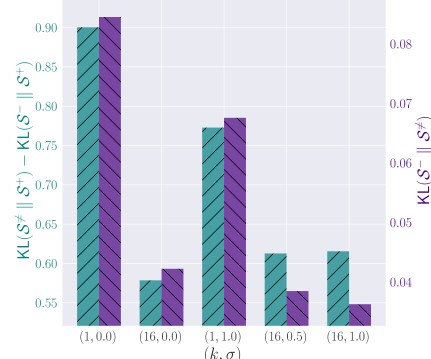

Figure 2: Effect of color jitter on the dis-
crepancy measures.

259 audio-video pairs from the same movie, which leads model to maintain less of the content-exclusive
260 artifacts in the embedding space. In the most extreme case, $k = 64$, all the training instances are

---

[7]Given a TV show, either all or none of its episodes are included in the held-out set.

sampled from the same movie. From Table 1, we observe that different variants of our sampling policy, with no imposed temporal constraint, *i.e.* $w = M_n$, outperform the baseline on all three downstream tasks

**Sampling window ($w$)** Smaller $w$ forces samples that belong to same movie to be drawn from a shorter temporal window, hence growing the probability that they look/sound very much alike (*i.e.* harder negative pairs). That is, it should further diminish the discrepancy measures. Figure 1b illustrates this behavior where we gradually increase $w$ while $k = 16$. However, from Table 1, it does not seem that tuning for $w$, *i.e* $w \neq M_n$, provides a meaningful gain on downstream tasks. This implies that commonalities which persist throughout the duration of a movie are sufficiently powerful signals to be exploited for generating hard negatives. We hypothesize that different *scenes* both within and across different movies and TV shows are of variety of length, thus a fixed $w$ is sub-optimal. Ideally, we should identify scene boundaries and and dynamically modify $w$ during sampling, something which we leave for future iterations of this work.

**Temporally adjacent samples.** Along the lines of previous observations, Figure 1c shows that indeed drawing temporally adjacent snippets from the same long-form content, *i.e.* $w = k$, results in aggressively reducing the discrepancy measures. This behavior is agnostic with respect to $k$ yet exacerbates as $k$ grows. Note that, the contrastive loss is an instance-discrimination objective function. Therefore, forcing it to distinguish between temporally adjacent snippets, that naturally sound and look extremely similar, leaves no choice for the model but to discard valuable semantic notions, which predictably leads to poor representations, also confirmed by result reported in Table 1.

Table 1: Ablation study of the proposed sampling policy on different downstream tasks, measured by top-1 classification accuracy.

| \multicolumn pretraining dataset: Movie | | | | |
|---|---|---|---|---|
| $k$ | $w$ | HMDB51 | ESC50 | UCF101 |
| 1 | – | 60.32 | 86.50 | 85.69 |
| 4 | $M_n$ | 61.37 | 89.91 | 85.38 |
| 8 | $M_n$ | 62.09 | 88.75 | 86.06 |
| 16 | $M_n$ | 62.92 | 88.33 | 86.30 |
| 32 | $M_n$ | 61.04 | 88.00 | 85.98 |
| 64 | $M_n$ | 61.30 | 86.83 | 85.43 |
| 16 | 64 | 60.26 | 87.00 | 83.61 |
| 16 | 128 | 60.58 | 86.50 | 85.30 |
| 16 | 256 | 62.02 | 87.75 | 84.85 |
| 16 | 512 | 61.30 | 87.08 | 85.38 |
| 16 | 1024 | 60.65 | 86.16 | 84.61 |
| 16 | 2048 | 61.83 | 87.66 | 85.11 |
| 4 | 4 | 60.19 | 88.00 | 84.66 |
| 16 | 16 | 56.86 | 88.75 | 82.71 |
| 64 | 64 | 57.45 | 84.58 | 82.68 |
| pretraining dataset: TV | | | | |
| $k$ | $w$ | HMDB51 | ESC50 | UCF101 |
| 1 | - | 56.40 | 85.50 | 84.37 |
| 8 | $M_n$ | 61.50 | 87.50 | 85.96 |
| 16 | $M_n$ | 61.69 | 89.00 | 85.64 |
| 8 | 64 | 60.58 | 88.00 | 85.96 |
| 8 | 128 | 60.00 | 85.66 | 85.77 |
| 16 | 256 | 61.30 | 86.41 | 85.01 |

**Movies vs. TV Shows.** To confirm that our sampling policy behaves consistently across both movies and TV shows, Figure 1d illustrates the discrepancy measures computed on TV dataset. We observe similar effectiveness when using $k$ and $w$ as tuning nobs for reducing either $\mathsf{KL}(\mathcal{S}^- \parallel \mathcal{S}^{\neq})$ or the gap between $\mathsf{KL}(\mathcal{S}^- \parallel \mathcal{S}^+)$ and $\mathsf{KL}(\mathcal{S}^{\neq} \parallel \mathcal{S}^+)$. Table 1 demonstrates that different variants of our approach significantly outperform the baseline, *i.e.* $k = 1$. We attribute the larger gains achieved when using TV instead of Movie dataset to the fact that content diversity is naturally lower when pretraining on TV shows since each one includes many episodes that all are characterized with the same content-exclusive artifacts.

**Color jitter.** We have established so far that commonalities which persist throughout the duration of a long-form content, things likely associated with color pallet, frequent appearance of the main character's face and voice, and repetitive scenes can be exploited for learning better representations. That is, one may naturally assume that employing data augmentation techniques like color jitter should be helpful since by distorting content-exclusive visual artifacts, color jitter is expected to reduce $\mathsf{KL}(\mathcal{S}^- \parallel \mathcal{S}^{\neq})$. Figure 2 illustrates the effect of color jitter, where brightness, contrast, and saturation jitter values are chosen uniformly from [max(0, $1 - \sigma$), $1 + \sigma$]. We observe that color jitter reduces the discrepancy measures for the baseline but not as much as it can be obtained by our proposed sampling policy ($k > 1$), and even then according to Table 2 only yields a slight gain on downstream tasks.

$\ell_2$**-normalized feature space.** The common practice [2, 31, 39, 7] is to compute contrastive loss in an $\ell_2$-normalized feature space, where according to [47] the temperature hyper-parameter, $\tau$, controls

Table 4: Effect of self-supervised learning from curated versus uncurated data on different downstream tasks. The "years" column indicates the duration of the pretraining datasets in years.

| method | pretraining dataset | uncurated | years | HMDB51 | ESC50 | UCF101 |
|--------|---------------------|-----------|-------|--------|-------|--------|
| Ours | Movie | ✓ | 0.7 | 62.9 | 88.3 | 86.3 |
| Ours | TV | ✓ | 0.7 | 61.7 | 89.0 | 85.6 |
| XDC[3] | IG-Random16M | ✓ | 5 | 55.2 | 84.3 | 84.1 |
| XDC[3] | IG-Random65M | ✓ | 21 | 61.2 | 86.3 | 88.8 |
| XDC[3] | IG-Kinetics16M | ✗ | 5 | 57.3 | 82.5 | 87.6 |
| XDC[3] | IG-Kinetics65M | ✗ | 21 | 63.1 | 84.8 | 91.5 |

the strength of penalties on hard negative samples. We explored this with two widely-used $\tau$ values. From Table 2, we observe that compared to operating in an unnormalized embedding space, adopting such design choice results in a large performance drop on HMDB51[23] while other downstream benchmarks see only negligible gains.

**Curated vs. Uncurated data.** To the best of our knowledge, the only other uncurated dataset used for audio-visual self-supervised learning is IG-Random[3][8]. Table 4 confirms that learning from uncurated movies and TV shows is extremely effective. Our results significantly exceed those of XDC[3] obtained on IG-Random16M despite using a simpler model and 7 times smaller volume of pretraining data. Even in comparison to IG-Random65M with 30 times larger data, we obtain better performances on 2 out of 3 benchmarks. The most promising of our findings though is how competitive our results are against XDC[3] when it is trained on variants of IG-Kinetics which are not only curated but also orders of magnitude larger. With all that, we confidently reject the notion that audio-visual self-supervised learning from uncurated data considerably lags behind utilizing large-scale curated datasets.

**Effect of genre.** The distribution of genre among movies used in our pretraining is the closest we have to taxonomy in the curated datasets. So, it is worth examining the quality of our learned representations under various genre distributions. To do so, given a fixed pretraining budget ($N =1.6$K), we compare four different scenarios where movies used in the pretraining are distributed i) non-uniformly over all genres except Drama, and Comedies, ii) non-uniformly over Drama, and Comedies, iii) uniformly over all genres, and iv) non-uniformly over all genres. Table 3 confirms that indeed there is very little difference between the aforementioned setups when it comes to transfer learning to the downstream tasks.

Table 2: Effect of color jitter ($\sigma$) and computing contrastive loss in $\ell_2$-normalized embedding space with temperature hyper-parameter ($\tau$) on different downstream tasks.

| $k$ | $\sigma$ | HMDB51 | ESC50 | UCF101 |
|-----|----------|--------|-------|--------|
| 1 | 0.0 | 60.32 | 86.50 | 85.69 |
| 16 | 0.0 | 62.92 | 88.33 | 86.30 |
| 1 | 1.0 | 60.45 | 87.66 | 84.82 |
| 16 | 0.5 | 60.13 | 87.75 | 85.98 |
| 16 | 1.0 | 61.11 | 88.33 | 85.93 |

| $k$ | $\tau$ | HMDB51 | ESC50 | UCF101 |
|-----|--------|--------|-------|--------|
| 16 | 0.07 | 60.78 | 87.08 | 86.86 |
| 16 | 0.30 | 60.78 | 89.25 | 85.72 |

Table 3: Effect of genre distribution in Movie dataset on different downstream tasks. Experiments are conducted with input spatial resolution of $112 \times 112$ pixels.

| setting | HMDB51 | ESC50 | UCF101 |
|---------|--------|-------|--------|
| i | 57.58 | 86.50 | 82.44 |
| ii | 56.99 | 85.50 | 82.39 |
| iii | 56.27 | 85.25 | 82.87 |
| iv | 56.40 | 86.75 | 83.24 |

### 4.3 Comparison with state-of-the-art

Table 5 compares our proposed approach of learning from Movies and TV shows against the best performing audio-visual self-supervised learning methods. In general, our numbers are comparable with the best existing results reported in the literature, even with much less data and considerably simpler model/training procedure[9]. It is interesting that training on Movie dataset alone obtains

---

[8]the data is not publicly available, and similarly the implementation to train XDC[3]

[9]supplemental material includes comparison of training costs

Table 5: Comparison with state-of-the-art. Dataset abbreviations: **A**udio**S**et[11], **H**ow**T**o100M[28], **IG**-Kinetics**65M** [12]; their length in years is given in the "years" column. "Arch." denotes the architecture of video backbone ($f$). [2]$^{\dagger}$ indicates when the corresponding model use only audio and video, and not text modality. For a fair comparison, when using only Movie dataset, we train for twice as many epochs as our other variants in order to match their total number of gradient updates.

| Method | Arch. | pretraining dataset | curated | years | HMDB51 | UCF101 | ESC50 |
|--------|-------|---------------------|---------|-------|--------|--------|-------|
| GDT[39] | R(2+1)D-18 | AS | ✓ | 1 | 66.1 | 92.5 | 88.5 |
| GDT[39] | R(2+1)D-18 | IG65M | ✓ | 21 | 72.8 | 95.2 | |
| XDC[3] | R(2+1)D-18 | AS | ✓ | 1 | 61.0 | 91.2 | 84.8 |
| XDC[3] | R(2+1)D-18 | IG65M | ✓ | 21 | 67.4 | 94.2 | |
| AVTS[22] | MC3 | AS | ✓ | 1 | 61.6 | 89.0 | 82.3 |
| AVID[31] | R(2+1)D-18 | AS | ✓ | 1 | 64.7 | 91.5 | 89.1 |
| MMV[2]$^{\dagger}$ | R(2+1)D-18 | AS | ✓ | 1 | 70.1 | 91.5 | 85.6 |
| MMV[2]$^{\dagger}$ | S3D-G | AS | ✓ | 1 | 68.2 | 90.1 | 86.1 |
| MMV[2]$^{\dagger}$ | S3D-G | AS+HT | ✓ | 16 | 68.3 | 91.1 | 87.2 |
| Ours ($k$=16) | R(2+1)D-18 | Movie | ✗ | 0.7 | 64.5 | 87.9 | 88.8 |
| Ours ($k$=8) | R(2+1)D-18 | Movie+TV | ✗ | 1.4 | 65.0 | 87.7 | 89.1 |
| Ours ($k$=16) | R(2+1)D-18 | Movie+TV | ✗ | 1.4 | 65.1 | 88.5 | 89.1 |
| Ours ($k$=32) | R(2+1)D-18 | Movie+TV | ✗ | 1.4 | 65.6 | 88.7 | 88.2 |

comparable performance to the cases where both TV and Movie datasets are used for pretraining. This further confirms the richness of the training data which movies and TV shows can provide to self-supervised learning problems. We also see that increasing $k$ even beyond 8 gives further incremental gains on action recognition benchmarks.

## 5   Conclusion

Despite its amazing recent progress, state-of-the-art self-supervised learning still heavily relies on supervised, *i.e.* curated, large-scale datasets for pretraining. In this work, we have shown that pretraining solely on uncurated data in forms of movies and TV shows, even at a comparatively small scale, can give rise to representations which are capable of competing with the state-of-the-art of more complex architectures trained on larger curated datasets. This comes contrary to the current literature which tends to suggest that learning from uncurated data largely falls behind the use of curated alternatives. We intentionally made design decisions to keep our approach and training strategy as simple as possible to demonstrate that learning decently powerful audio-visual representations does not necessarily require gigantic data and compute resources. Through extensive set of experiments, our work establishes for the first time the efficacy of self-supervised learning of audio-visual representations from movies and TV shows.

## 6   Broader impact

**Potential benefits.** Our work shows that competitive multimodal representations can be learned from a comparatively small volume of *uncurated* data in the form of movies and TV shows. Besides minimizing any sort of human-involvement, which we believe must have already been paid an extra attention to in the literature, our work demonstrates that one does not require gigantic data and compute resources for effective self-supervised pretraining. Such results promise a more democratized research arena where smaller groups are not alienated due lack of sufficient compute resources. More importantly, lowering the compute requirements naturally reduces any environmental effects which training these models can potentially have.

**Potential risks.** Any machine learning method is susceptible to the potential underlying biases in the data. This is more important for self-supervised methods that deal with huge volumes, often not evaluated by diverse group of humans for any fairness concerns. The same is generally true in our case which requires us to make sure that titles that are included in training are diverse and inclusive.

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
