# OpenReview forum: "Watching Too Much Television is Good: Self-Supervised Audio-Visual Representation Learning from Movies and TV Shows"
_NeurIPS.cc/2021/Conference — NeurIPS 2021 Submitted_

### Official Review · Reviewer_zZqb · 2021-07-08

**Rating:** 4
**Confidence:** 4

**Summary:**

This paper performs self-supervised, audio-visual pretraining of deep CNNs on movies and TV shows. The downstream tasks investigated are action classification in videos (HMDB51 and UCF101 datasets) and environmental audio classification (ESC50 dataset). The paper shows that pre-training on 0.7-1.4 years worth of TV and movies results in downstream task performance that is competitive with, but lags slightly behind, the current state-of-the-art approaches that pre-train on datasets like HowTo100M, AudioSet, and IG-Kinetics65M.

**Limitations And Societal Impact:**

The authors do acknowledge the copyright restrictions which would make their dataset generally inaccessible to most of the research community.

**Main Review:**

Strengths:

- The paper studies a timely problem that is of broad and growing interest to the community

- The paper is clear and easy to read

- The paper demonstrates that Hollywood Movies and TV shows are an effective data source for learning audio-visual representations


Weaknesses:

- Aside from the novel dataset used, the paper offers little in the way of modeling or algorithmic novelty over the existing state of the art.

- The downstream results slightly lag current SotA approaches trained on other widely-used datasets

- It would have been better to also run baseline models on the movie dataset, as current experiments tangle up architecture with training data

- One of the paper's main arguments is that uncurated data is preferable over curated datasets, but the licensing and distribution restrictions surrounding the data used here (movies, TV shows) make it even more difficult for the research community to use than existing datasets based on e.g. YouTube. It’s also not clear to me why a collection of Hollywood movies and TV shows that are categorized by genre qualifies as “uncurated” when a dataset of how-to videos scraped from youtube (e.g. HowTo100M) does not.


Suggestions to authors:

It seems to me that the experiments here show that long-form content like movies offer no advantage over scraped youtube videos when incorporated into architectures that make i.i.d. assumptions about the relationship of clips drawn from the same video. It seems like a key difference between long-form content and youtube content is the presence of an extensive overarching narrative structure, which I can see being useful in future work but wasn’t leveraged here.

I would have liked to seen a comparison between the TV+Movie dataset and a subset of HT100M or IG-Kinetics65M that was sized to match the duration of the TV+Movie dataset; the current results don’t establish why pre-training on TV+Movies lags behind the other datasets. Is it simply because the other datasets are larger, or is one video domain naturally more informative than others?

Following from the previous point, it would have been even more informative if the paper included experiments that held the modeling architecture and learning algorithm fixed, and varied the dataset used for pre-training.

-----Post rebuttal update:
After reading the authors' feedback and the thoughts of the other reviewers, my assessment of the paper remains unchanged.


**Time Spent Reviewing:**

2.5

---

> ### Author Response · Authors · 2021-08-09
> **Response to reviewer zZqb**
>
> We thank the reviewer for the valuable feedback. Next, we’ll address them to the best of our ability.
>
> $\textbf{``Algorithmic Novelty'' }$
>
> The objective of this paper is not proposing novel datasets or algorithms. Instead the paper invites readers to rethink the implicit supervision that exists when a supervised and curated dataset is used in a self-supervised fashion which is the common practice in the literature. That is why we intentionally made fairly standard design choices including basic NCE loss function. We also demonstrate that unlike earlier observations, one can learn very competitive self-supervised models from the uncurated data at a comparable scale. To the best of our knowledge, none of these two objectives have been comprehensively addressed in the literature before.
>
> $\textbf{``Baseline models on the movie dataset" }$
>
> The feedback is not very clear so we assume by “baseline models” the reviewer means other SOTA algorithms reported in Table 5. At the time of submission, GDT, XDC (refer to footnote 8) and MMV did not have code available for training. Also MMV formulation, when using audio and video, boils down to our own objective function.
>
> $\textbf{``Uncurated data and distribution restrictions" }$
>
> We agree with the reviewer that in terms of accessibility, using Movies and TV shows is more restrictive than videos scrapped from YouTube. However, the point is that none of the previous works simply and blindly scrapped YouTube to create their training data. They used supervised, curated and well-trimmed datasets. The only exception is IG-Random (for the sake of argument let’s assume YouTube ~ Instagram) against which we have extensively compared. As for the HowTo100M dataset there are a few points to mention. First, referring to Section 3 of the HowTo100M paper shows that indeed there has been a lot of curation happening throughout the data collection process (e.g making sure only “visual tasks” are included, etc.) so it is far from blindly downloading videos from YouTube. Note that in Movies there are scenes where audio does not correlate with the visuals like the background music or a dialogue yet we did not exclude such scenes rather kept the data untouched. Second, the audio is removed from videos hosted on the HowTo100M download page. Finally, Table 3 shows that our model is agnostic with respect to the effect of genre.
>
> $\textbf{``Comparison with a subset of HT100M or IG-Kinetics65M" }$
>
> We would like to draw the reviewer's attention to the fact that data hosted on HowTo100M does not have audio, it is simply video and text. Furthermore, as also mentioned in the paper IG-Kinetics is not publicly available so we are incapable of conducting the suggested experiments.
>
> $\textbf{``Fixed algorithm with varying dataset used for pre-training" }$
>
> The focus of this paper is not on algorithmic novelty rather drawing attention to the implicit effects of utilizing curated datasets in self-supervised learning. As was also mentioned in the paper, the only other uncurated dataset is IG-Random which is not publicly available.

---

### Official Review · Reviewer_Lidi · 2021-07-17

**Rating:** 4
**Confidence:** 4

**Summary:**

The task is to learn audio-visual representations from uncurated data such as movies and TV videos. Unlike previous works that show learning from uncurated data is ineffective, the authors show good performance from learning from these videos. They propose hierachical sampling policy, with negatives being sampled from both within and across different sources. The model is trained on the dataset that the authors collected, which consists of thousands of hours. The performance on various downstream tasks (action recognition, sound classification) is on par compared to using much larger uncurated datasets such as IG-Random.

**Ethical Concerns:**

No concerns

**Limitations And Societal Impact:**

The explanation in Sec 6 is adequate.

**Main Review:**

The authors tackle an important problem of learning uncurated datasets. The effective use of uncurated data is an important question and a future direction for the ML community.

The dataset used in the paper is of significant size -- 1.4 years (12k hours). The sources are from TV/ movies, which is already somewhat refined and curated compared to user-created content such as IG-Random. This might explain the performance advantage over IG-Random. The data has not been released due to copyright.

The method used is fairly standard contrastive learning. The authors propose sampling strategies to adjust the difficulty of negatives from same and different source videos. The taking of negatives from same/different videos has been shown in previous works, but the adjustment of temporal windows based on the intuition on formulation on scenes is original. However, the effect of sampling strategies shown in Table 1 is marginal, which leads me to think that the good performance comes from the quality of inputs rather than the training strategy.

Overall, it is notable that the authors highlighted the TV/ movies data as a good source of self-supervised learning, but the technical contributions are not sufficient to merit presentation at the conference.

The writing is clear.

**Time Spent Reviewing:**

2hr

---

> ### Author Response · Authors · 2021-08-09
> **Response to reviewer Lidi**
>
> We thank the reviewer for the valuable feedback. The choice of a fairly standard contrastive loss function was intentional. Please note that the objective of this paper is not proposing a novel dataset or an algorithm. Instead the paper invites readers to rethink the implicit supervision that exists when a supervised and curated dataset is used in a self-supervised fashion. We also demonstrate that unlike earlier observations, one can learn very competitive self-supervised models from uncurated data at a comparable scale. To the best of our knowledge, none of these two objectives have been comprehensively addressed in the literature before. It is worth noting that earlier works which sample negatives from the same instance use very short videos often in orders of 10 seconds to at most a few minutes in case of HowTo100M. In our case, each movie is on average about 1.5 hours long so the operating regime is quite different and so is our KL-based analysis to the best of our knowledge.

---

> > ### Comment · Reviewer_Lidi · 2021-08-26
> > **Final comments**
> >
> > Thanks for the comments. I understand the value of the paper is in the use of the uncurated data, but I still maintain the view that the contribution is not enough to merit publication at Neurips.

---

### Official Review · Reviewer_cgAj · 2021-07-17

**Rating:** 4
**Confidence:** 4

**Summary:**

The paper proposes a method to learn self-supervised audio-visual representations from long (order of half to couple hours) uncurated videos namely from movies and TV shows. A new sampling approach is presented to ensure batches are created while taking into consideration the effect of re-occurring audio-visual patterns (main character or background scenes). Picking multiple clips from the same long video that are spaced by some minimum distance is found to be effective as compared against random sampling. Pretrained models are evaluated on standard action recognition tasks: UCF101 and HMDB, and audio classification on ESC50.

**Ethical Concerns:**

Perhaps some, related to the use of copyrighted materials -- movies and TV series -- for training. The paper does talk about it and indicates that there is precedence for it, however, does not mean that previous works had obtained permission.

**Limitations And Societal Impact:**

Some worry of using the models trained on movies for downstream tasks as movies may often be racially insensitive and surface social problems. This is however indicated in the paper.

**Main Review:**

*Strengths:*
1. The paper makes a strong case for using uncurated datasets used for pretraining.
2. Using KL divergence to measure the quality of batches created for contrastive learning is interesting. In particular, desirable features are observed by the proposed sampling scheme - a low KLD between negative clips from the same movie ($S^-$) and other movies ($S^\neq$), and a small difference between $S^\neq$ and $S^+$ and $S^-$ and $S^+$ (see Fig. 1).

*Weaknesses:*
1. Learning from uncurated data. Firstly, I'm not convinced that movies and TV shows are a uncurated dataset. In fact, as compared to most user generated content on Youtube, one could argue that they are highly curated with a large amount of editing and production value. Please also see point 2, which clarifies how movie data is not uncurated with respect to downstream tasks either.

*Discussion for point 1, not influencing the rating*. I think the paper raises some interesting questions on whether it is possible to truly acquire random uncurated audio-visual data. Randomly picking Instagram/Youtube ids too has a bias of the video being on Instagram/Youtube in the first place, so it's hard to say that IG-Random [3] is in fact uncurated. In fact, I'm not even convinced that we need to train machines on uncurated videos - humans don't simply listen and watch *everything* to learn about the world, we go study at schools and universities and experience curated learning.

2. One of the downstream evaluation datasets, HMDB51, has a majority of clips sourced from movies. HMDB51 also tends to show highest performance improvements for the video representation in the experiments when training with movies (+2.6% in Table 1 over k=1), while UCF101 which features the sports categories, not covered commonly in movies and TV shows, does not improve much (+0.6% in Table 1). This raises the question of whether we should be training on uncurated data. More importantly it goes against one of the claims on lines 72-73: that there is a "meaningful domain gap between our pretraining data and space of downstream tasks". In fact, ESC50 categories may also benefit more by sound effects inserted in TV/movies as compared to speech or background-music heavy vlogs, messages, or instructional videos found on youtube/instagram, but this would require further investigation.

3. While the core idea presented here has merit - how to sample when given long videos - I don't see anything special catered towards movies/TV, for example, by actually analyzing their content. I see two options:
a) Either position the paper on how to create batches from long videos and also demonstrate the effectiveness of the sampling algorithm by pretraining on other datasets with long(ish) videos such as HowTo100M (million videos of 6 minutes average duration, admittedly not hours); or
b) it would be useful to incorporate some movie/TV content analysis if the core message is about the benefits of training on such data.

4. (not influencing rating) The argument about such data democratizing research for smaller groups with limited compute resources should be removed. Firstly, such data is not commonly available: 3600 films and 9200 TV episodes. Secondly, training required almost 2 days with 8 A100 GPUs - 320 GB memory could often be 50-100% of the GPU memory of an academic lab, 32 GPUs x 10GB. In such a case, even creating just the final version of Fig. 1 (20+ experiments), will require all GPUs used for 40 days!

5. Minor:
a) Fig. 1b, k is said to be 16. Similarly, what is w for Fig. 1a?
b) Is the batch size 64 as indicated on L260?
c) Making highest performance numbers bold in the tables would improve readability.
d) While Fig. 1 is very interesting, it would also be useful to include the actual values of KL(S^- || S^+) as an indicator of training performance, since the desired low scores can be achieved for the other metrics with a failure mode in training where all S+, S-, S^\neq lead to similar scores.



**Time Spent Reviewing:**

4

---

> ### Author Response · Authors · 2021-08-09
> **Response to reviewer cgAj**
>
> We thank the reviewer for the valuable feedback. Next, we’ll address them to the best of our ability.
>
> $\textbf{``Learning from uncurated data''}$
>
> We agree with the reviewer that from an aesthetic point of view (e.g no camera instability or low quality audio), movies are indeed curated however this is different from utilizing supervised and well-trimmed video datasets which are widely employed in literature for self-supervised learning (ref. Lines 38-55). We would have compared the effect of training on movies versus IG-Random yet unfortunately the IG-Random dataset is not publicly available and similarly the implementation in order to train XDC on our data.
>
> $\textbf{``On downstream evaluation datasets''}$
>
> Solely looking at the improvement margin, especially when two baselines are dramatically different is not conclusive. UCF101 is approximately twice larger than HMDB51 (hence fine-tuning has a better chance to compensate for differences in pre-training) with a 20-25% (based on different models) higher baseline. So, it is only expected that getting further improvements on UCF101 will be a harder task. In regards to the domain gap, while speculating on the similarity of content will result in subjective assessments which we agree with the reviewer that they need further investigation, similarity and relations between the label set of curated pre-training and downstream data is pretty clear and even pointed out by XDC authors too. As for HMDB51 being created mostly from movies, we could not find the name of those movies on HMDB51’s web page to cross reference with those used in our pre-training. However, HMDB51 was published in 2011 and less than 14% of movies used in our training were produced before 2011. Hence it is a very small likelihood that we share meaningfully large numbers of movies, if any such that it influences our conclusions.
>
> $\textbf{``Nothing special catered towards movies/TV''}$
>
> The objective of this paper is not proposing a novel algorithm, even though in the spirit of completeness we performed an extensive set of experiments to analyze the sampling policy. The paper simply invites readers to rethink the implicit supervision that exists when a supervised and curated dataset is used in a self-supervised fashion which is the common practice. On suggestion "a", please note that audio has been removed from the HowTo100M videos hosted on the dataset repo page. As for suggestion "b", the core message of the paper is on the importance of using uncurated data and Movies and TV shows were a source of that to which we had access in abundance.
>
> $\textbf{``Minor'}$
>
> a) It is M_n. Please refer to the first block of Table 1.
>
> b) Total batch size is 512 but divided on 8 GPUs results in 64 on each.

---

> > ### Comment · Reviewer_cgAj · 2021-08-30
> > **Thanks**
> >
> > Thanks to the authors for their detailed response. I think this concern has been raised by other reviewers as well - calling a large collection of movies / TV series as uncurated remains a sticky point for me.
> >
> > I appreciate the honest comment on another review
> >
> > > We happen to have access to movies and TV shows at scale.
> >
> > and therefore this work is a
> >
> > > call out to rethinking how supervised and curated data is used and viewed within the self-supervised learning literature
> >
> > and think that this is exactly how the paper should have been positioned. An exploratory call for how movies/TV series (long videos) may be used for self-supervised learning, without emphasizing the curated vs. uncurated positions too much. To add more to the contribution, it would be meaningful to do something that helps improve learning from such high quality edited videos.
> >
> > Regarding the UCF101 vs. HMDB51. I agree that UCF has twice as many labels as HMDB and therefore it's not as straightforward. However, just based on the nature of the label space (sports heavy UCF vs. movie actions HMDB) it's fair to assume that using movies helped improve performance for HMDB like datasets. BTW, there's nothing wrong in that and it happens for all self-supervised learning works. The issue is however with positioning the contribution with strong statements such as
> >
> > > meaningful domain gap between our pretraining data and the space of downstream tasks
> >
> > which I don't think is true.
> >
> > Considering other reviewer opinions as well, I agree that this may not be a sufficient contribution for NeurIPS and would keep my rating.

---

### Official Review · Reviewer_8Arr · 2021-07-21

**Rating:** 5
**Confidence:** 5

**Summary:**

The paper works on self-supervised audio-visual representation learning. It shows uncurated movies and TV shows are good sources for learning audio-visual representation. The proposed model targets on short video clips like 3-second clips, and learned representation is evaluated on action classification and audio classification.

**Ethical Concerns:**

There is no major ethical concern.

**Limitations And Societal Impact:**

Given the fact that the TV and movie datasets cannot be released and only the pretrained models will be publicly shared, I think the representation quality is not strong enough (measured by downstream performance). If the TV and movie dataset can be released it would be an impactful dataset for the community, but without that the technical contribution of the paper is limited.

There are some properties of the TV and movie datasets that are neglected. I think the audio accompanying TVs and movies could be background music (not much relevant to visual information), human’s speech (may correspond to the human's face/body), and the sound of objects/scenes that the model aims to learn. In other words, some portion of the audio is noise for audio-visual representation learning. But the proposed basic contrastive loss does not explicitly handle this.

The tech details about sampling policy (e.g. sampling clips from the same source video as hard negative, and the sampling windows affects difficulties) was studied in previous audio-visual paper [1] and video paper [2]. Thus the novelty is limited.

Besides, I think the experiments do not match the claimed contribution very well. It feels the claimed main contribution is the power of large-scale TV and movie datasets, then maybe the paper can consider comparing the same method trained on other TV/movie datasets (like [3]).

* [1] Cooperative Learning of Audio and Video Models from Self-Supervised Synchronization, Korbar et al.
* [2] Video Representation Learning by Dense Predictive Coding, Han et al.
* [3] Condensed Movies: Story Based Retrieval with Contextual Embeddings, Bain et al.


**Main Review:**

The paper trains models on a large-scale TV and movie dataset for representation learning, and it shows uncurated TVs and movies are good for audio-visual representation learning with a simple contrastive method. This message is valuable because there are not many large-scale TV/movie datasets available for representation learning. But the technical contribution of the paper is limited. The key method is a basic contrastive loss, and additionally, the paper explores sampling strategy, which is not very new (see limitation).

The paper provides detailed experimental studies on sampling strategy, color jittering and the effect of genre. The quality of submission is good.

Overall, the paper is clearly written. But figure 1 is hard to read. I suggest giving short names for different KL divergence values and adding more captions.

The key message from the paper is encouraging, showing that uncurated TV shows and movies are a good data source for audio-visual representation learning. However, public access to these copy-right datasets is a long-standing problem. The result of the paper will be more significant if these datasets can be publicly released in some way.


**Time Spent Reviewing:**

4

---

> ### Author Response · Authors · 2021-08-10
> **Response to reviewer 8Arr**
>
> We thank the reviewer for the valuable feedback. Next, we’ll address them to the best of our ability.
>
> $\textbf{``Given the fact that the TV and movie datasets cannot be released ...''}$
>
> It is not quite clear what exactly do the restrictions on publicly releasing the dataset have to do with the quality of learned representations that are measured on downstream tasks!
>
> $\textbf{``Limited contribution''}$
>
> The choices of a basic contrastive loss function or simple sampling policy were intentional. Please note that the objective of this paper is not proposing a novel dataset or an algorithm. The paper simply invites readers to rethink the implicit supervision that exists when a supervised and curated dataset is used in a self-supervised fashion. We also demonstrate that unlike earlier observations (XDC on IG-Random), one can learn very competitive self-supervised models from uncurated data at a comparable scale. None of these two objectives have been addressed comprehensively in the literature before. The core message of the paper is on the importance of using uncurated data and Movies and TV shows were a source of that to which we had access in abundance. We like to draw the reviewer’s attention to the fact that using proprietary data for training while publicly releasing pretrained models has many precedents; a number of them were noted in the Line 214.
>
> $\textbf{``Neglected properties of the TV and movie datasets''}$
>
> While we agree with the reviewer on potential improvements that could be made by refining the data as suggested, we like to kindly remind the reviewer that, the same as fairly standard choice of basic contrastive loss function, here we minimized any refinement of the data to show that even such simple framework using uncurated data can give rise to very competitive representations. This observation is the core message of the paper and goes against the current narrative in the literature which focuses on using curated data.
>
> $\textbf{``Novelty of sampling policy''}$
>
> Please note that earlier works which sample negatives from the same instance use very short videos often in orders of 10 seconds to at most a few minutes. In our case, each movie is on average about 1.5 hours long so the operating regime is quite different and so is our KL-based analysis to the best of our knowledge. Furthermore, we never claimed to be the first who proposed sampling negatives from the same video, we simply adopted it in the context of long-form content to improve the quality of learned representations.
>
> $\textbf{``Argument on the main contribution''}$
>
> The core message of the paper is NOT about the power of large-scale TV/Movie datasets. As mentioned before it is a call out to rethinking how supervised and curated data is used and viewed within the self-supervised learning literature. We happen to have access to movies and TV shows at scale. The other alternative for such study of uncurated data is user generated content which was already explored by IG-Random in XDC paper and we compared against it extensively.

---

> > ### Comment · Reviewer_8Arr · 2021-08-25
> > **Comments after the rebuttal**
> >
> > Thank authors for providing rebuttal. I read all the reviews and authors' rebuttals.
> > The rebuttal doesn't change my opinion that the technical contribution is not strong enough.
> > As a suggestion, I agree with reviewer cgAj (weakness point 3) that positioning on long video processing or showing the specific benefits of TV/movie w.r.t. other long videos would be interesting.
> > I will give 'rejection' as my final rating.

---

### Decision · Program_Chairs · 2021-09-28

**Decision:**

Reject

**Comment:**

The paper has a good message about using uncurated data for audio-visual self-supervised learning. However, the reviewers were not convinced that this paper carried through on the message, since TV material and movies are curated to an extent (though not class balanced).

The reviewers identified a number of weaknesses, including:

- insufficient novelty

- algorithm not applied to other (less curated) datasets

The rebuttal did not change their opinion.

The authors should take note of the suggestions for how to improve the paper,  particularly from Reviewer cgAj


**Consistency Experiment:**

NeurIPS has a long history of experimentation. In 2014, NeurIPS ran an experiment in which 10% of submissions were reviewed by two independent committees to quantify the randomness in the review process. This year, we repeated a variant of this experiment to see how the quality of the review process has changed over time.  This paper was part of the experiment and was therefore assigned to two committees (consisting of reviewers, an Area Chair, and a Senior Area Chair) that reached independent decisions.  If both committees made the same recommendation, this recommendation was followed. If a single committee recommended acceptance, the paper was accepted (with the exception of a few cases in which the other committee identified what we considered a fatal flaw, e.g., an error in a key result).

Both committees reached the same decision: **Reject**

The other committee assigned to the paper recommended **Reject**.  You can find the other set of reviews, along with any follow up discussion with the authors here:
https://openreview.net/forum?id=xZ6UeJ54wD